# Serum *P*-Cresyl Sulfate Is a Predictor of Central Arterial Stiffness in Patients on Maintenance Hemodialysis

**DOI:** 10.3390/toxins12010010

**Published:** 2019-12-21

**Authors:** Yu-Hsien Lai, Chih-Hsien Wang, Chiu-Huang Kuo, Yu-Li Lin, Jen-Pi Tsai, Bang-Gee Hsu

**Affiliations:** 1School of Medicine, Tzu Chi University, Hualien 97004, Taiwan; hsienhsien@gmail.com (Y.-H.L.); wangch33@gmail.com (C.-H.W.); nomo8931126@gmail.com (Y.-L.L.); 2Division of Nephrology, Hualien Tzu Chi Hospital, Buddhist Tzu Chi Medical Foundation, Hualien 97004, Taiwan; 3Division of Nephrology, Department of Internal Medicine, Dalin Tzu Chi Hospital, Buddhist Tzu Chi Medical Foundation, Chiayi 62247, Taiwan

**Keywords:** arterial stiffness, carotid–femoral pulse wave velocity, hemodialysis, *p*-cresyl sulfate

## Abstract

Arterial stiffness (AS) has an important impact on the outcomes of patients on hemodialysis (HD), and *p*-cresyl sulfate (PC) can mediate the process of vascular damage. We aimed to investigate the relationship between carotid–femoral pulse wave velocity (cfPWV) and the level of PCs in HD patients. Serum PCs were quantified using liquid chromatography mass spectrometry. Patients who were on standard HD for more than 3 months were enrolled and categorized according to the cfPWV into the high AS (>10 m/s) and control (≤10 m/s) groups. Forty-nine (41.5%) patients belonged to the high AS group and had a higher incidence of diabetes mellitus (DM) and increased systolic blood pressure, serum C-reactive protein, and PC levels but had lower creatinine, compared with those in the control group. In HD patients, the risk for developing high AS increased in the presence of DM (OR 4.147, 95% confidence interval (CI) 1.497–11.491) and high PCs (OR 1.067, 95% CI 1.002–1.136). Having DM (r = 0.446) and high PC level (r = 0.174) were positively associated with cfPWV. The most optimal cutoff value of PC for predicting AS was 18.99 mg/L (area under the curve 0.661, 95% CI 0.568–0.746). We concluded that DM and PCs were promising predictors of high AS in patients on maintenance HD.

## 1. Introduction

Arterial stiffness (AS), which is a risk factor for cardiovascular (CV) disease (CVD), has long been recognized as the main cause of mortality in patients with chronic kidney disease (CKD) [1,2,3]. Along with the other traditional risk factors, such as age, hypertension (HTN), diabetes mellitus (DM), and uremia-related factors, including inflammation and abnormal bone and mineral metabolism, vascular calcification has been well known to be associated with CVD [4]. Increasing evidence has shown that the process of vascular damage is mediated by proteins, such as alkaline phosphatase, fetuin A, and parathyroid hormone; abnormal calcium and phosphate homeostasis; and protein-bound uremic toxins [5,6]. Pulse wave velocity (PWV) is a noninvasive method to measure vascular function and has been regarded as a strong predictor of CV events and mortality in patients with end-stage renal disease (ESRD), independent of the classical CV risk factors [2,3]. In the Chronic Renal Insufficiency Cohort (CRIC) study, CKD patients with high PWV were shown to be more likely to develop adverse renal outcomes, including decrease in renal function by half, ESRD, or death [7].

*P*-cresyl sulfate (PC), which is a 188-kDa gut-derived and protein-bound uremic toxin, was shown to progressively accumulate as renal function worsened [8] and has been regarded as a detrimental factor for renal fibrosis by enhancing the production of reactive oxygen species and by activating transforming growth factor β and the renal–angiotensin–aldosterone system [9,10]. Furthermore, PC has been linked with endothelial dysfunction in vitro [11,12], AS, vascular calcification [13], CV events, and even all-cause mortality in patients with CKD and on hemodialysis (HD) [13,14,15].

Given the aforementioned data, PWV could predict CV morbidity and mortality and PCs could play a role in AS and lead to adverse outcomes. However, the relationship between PCs and carotid–femoral PWV (cfPWV) in a CKD population is unknown. We conducted this study to noninvasively measure vascular function using cfPWV and to examine the possible risk factors, especially serum PC levels, for developing AS in patients on HD.

## 2. Results

Of all the HD patients, 59 (50%) were women, the mean age was 63.05 ± 13.28 years, and the median duration of receiving HD was 56.02 months (interquartile range (IQR) 24.6–111.45 months); 55 (46.6%) and 66 (55.9%) patients had DM and HTN, respectively. As measures of adequacy of dialysis, the mean Kt/V was 1.35 ± 0.17, and the mean urea reduction ratio was 0.74 ± 0.04. The mean total PC level of all HD patients was 16.57 ± 9.07 mg/L (Table 1).

Forty-nine patients (41.5%) were diagnosed as high central AS. Compared with the control group, the high central AS group had higher percentage of DM (73.5% vs. 27.5%, *p* < 0.001); higher systolic blood pressure (SBP; 147.84 ± 23.38 mmHg vs. 138.07 ± 26.99 mmHg, *p* = 0.021); higher serum levels of glucose (143.00 (IQR 119.50–206.00) mg/dL vs. 132.00 (IQR 110.50–162.00) mg/dL, *p* = 0.042); higher C-reactive protein (CRP; 0.57 (IQR 0.13–1.09) vs. 0.24 (IQR 0.08–0.86) mg/dL, *p* = 0.029); and higher total PC levels (20.26 ± 11.27 mg/dL vs. 13.95 ± 5.93 mg/dL, *p* = 0.047) but had lower levels of creatinine (8.73 ± 1.94 mg/L vs. 9.47 ± 1.98 mg/L, *p* < 0.001) (Table 1). There were no significant differences in HTN prevalence, HD duration, body composition, HD adequacy, lipid profiles, and the other clinical characteristics or medications between these two groups.

After adjusting for various factors, including overall age, sex, HD duration, PC levels, DM, SBP, heart rate, CRP, glucose, and creatinine, multivariate logistic regression analysis showed that total PC levels (adjusted odds ratio (aOR) 1.072, 95% confidence interval (CI) 1.002–1.147, *p* = 0.043) and DM (aOR 4.095, 95% CI 1.429–11.739, *p* = 0.009) were the significant independent risk factors for developing high AS (Table 2).

Simple linear regression analysis showed that the value of cfPWV was significantly positively correlated with DM, SBP, logarithmically transformed glucose, and total PC levels, but was negatively correlated with the logarithmically transformed HD duration (Table 3). On multivariate stepwise linear regression analysis, DM (r = 0.446, *p* < 0.001) and PC levels (r = 0.174, *p* = 0.018) had significant positive correlations with cfPWV.

Receiver-operating characteristic (ROC) curve analysis (Figure 1) showed that the best cutoff serum level of PC to predict high AS in HD patients was 18.99 mg/L with area under the curve (AUC) of 0.661 (95% CI 0.568–0.746, *p* = 0.002), sensitivity of 48.98% (95% CI 34.4% to 63.7%), and specificity of 84.06% (95% CI 73.3% to 91.8%).

## 3. Discussion

This study showed that DM and high serum PC levels were associated with high cfPWV and could be predictors of high AS in patients on HD. In patients with decline in renal function, AS has been well known to cross-talk with CV events and could lead to poor long-term outcomes in CKD and ESRD patients [2,3,16]. AS that is caused by vascular calcification, which is secondary to an imbalance between the inhibitors and promoters of vascular osteogenesis, and by the traditional and CKD-related risk factors has been reported to progressively increase as renal function declines [5,17]. Vlachopoulos et al. showed that vascular function, which was noninvasively measured and presented as PWV, had a linear correlation with the pooled relative risks for CV events and mortality [3]. Risk factors, such as DM and HTN, had been shown to be related with AS, as measured by cfPWV [18]. Moreover, evidence has shown that deteriorating glucose tolerance was independently associated with central AS with decreasing arterial compliance, carotid–femoral transit time, and increased aortic augmentation index [19]. Furthermore, Agnoletti et al. demonstrated that longer duration of DM led to a higher cfPWV, independent of the other risk factors for AS [20]. In a healthy population, McEniery et al. showed that aortic PWV was associated with higher computed tomography-proven calcification score and isolated systolic HTN [21]. Together with these studies, Cecelja et al. conducted a systemic review and reported that PWV was associated with old age, BP, and DM [18]. Inflammation has been postulated to be associated with endothelial dysfunction and AS. However, the CRIC study showed that baseline inflammation could not predict the long-term AS changes, although there was a positive correlation between several inflammatory markers and AS; these findings highlighted that there were other factors more important than inflammation that cause AS in patients with CKD [22]. Taken together, we found that HD patients with high AS had higher prevalence of DM, SBP, CRP levels, and degree of cfPWV. Similarly, after adjusting for the confounders, we found that DM was the independently significant predictor for the development of high AS in patients on HD.

Initially being known as a gut-derived and protein-bound uremic toxin, PC levels have been shown to increase and accumulate as renal function declined and led to the progression of renal dysfunction and all-cause mortality in patients with CKD [8]. In one systemic review, PC was found to activate oxidative stress, enhance cytokine and inflammatory genes, and induce renal tubular damage [6]. In addition to being regarded as a detrimental factor for renal fibrosis through enhancement of the production of reactive oxygen species, activating transforming growth factor β and the renal–angiotensin–aldosterone system [9,10], PC levels have been reported by in vitro and human studies to induce endothelial dysfunction by increasing the number of circulating endothelial microparticles [23]. Furthermore, an in vitro study on human umbilical vein endothelial cells revealed that PCs could contribute to endothelial dysfunction through the mechanism of increased endothelial permeability, along with reorganized presentation of endothelial actin and VE cadherin and inhibition of endothelial proliferation and wound repair in a dose-dependent manner [11,12]. In addition to playing a role in the progression of endothelial dysfunction, PCs were found to be correlated with image-proven vascular calcification and cfPWV, together with an inverse relationship with the estimated glomerular filtration rate of CKD patients [13]. Recently, Opdebeeck et al. proved that short- and long-term exposures to PCs promoted aortic inflammation and calcification, respectively, in vivo through the acute-phase response and coagulation signaling pathway [24]. In a cross-sectional study, Rossi et al. reported that serum PC was independently associated with interleukin 6 and PWV, highlighting its role in inflammation and its contribution to CV damages in CKD stages 3–4 [25]. Some cohort studies showed that beyond the traditional risk factors, such as age, DM, CRP, malnutrition, or Framingham risk scores, PC levels and severity of vascular calcification led to higher CV events and mortality in CKD and HD patients [13,14,15]. In accordance with these studies, we found that PCs correlated positively with cfPWV and could be regarded as a main risk factor for developing AS in patients on HD.

A limitation of this study was its cross-sectional and single-center design and the limited number of HD patients. Therefore, the causal relationship between serum PC levels and central AS in patients on HD should be investigated in longitudinal studies on a larger number of patients.

## 4. Conclusions

Together with DM, serum PC level of >18.99 mg/L may be a risk factor and a predictor of the development of AS in patients on HD. These findings indicated that the gut-derived uremic toxin PC might mediate the process of AS, but its definite mechanism needs to be further elucidated.

## 5. Materials and Methods

### 5.1. Participants

From October 2017 to February 2018, 118 patients on HD at a medical center were enrolled. The inclusion criteria were age older than 20 years and receipt of standard 4-h HD three times per week for at least 3 months using standard bicarbonate dialysate and disposable high flux polysulfone artificial kidney (FX class dialyzer, Fresenius Medical Care, Bad Homburg, Germany). The exclusion criteria were active infection, acute myocardial infarction, stroke, peripheral arterial occlusive disease, pulmonary edema, or refusal to provide informed consent. This study was approved by the Protection of Human Subjects Institutional Review Board of Tzu Chi University and Hospital (IRB 103-136-B and 108-96-B).

### 5.2. Biochemical and Anthropometic Analyses

After an HD session, the body weight and height were measured to the nearest half kilogram and half centimeter, respectively, with the patients wearing light clothing. Body mass index was calculated as the weight (kg)/height (m)^2^.

Before HD, approximately 5 mL of blood was collected from each patient. This blood sample was centrifuged at 3000× *g* for 10 min, stored at 4 °C, and used for biochemical analyses within 1 h after collection. The serum levels of blood urea nitrogen, creatinine, glucose, total cholesterol, triglyceride, total calcium, and phosphorus were examined by an autoanalyzer (Siemens Advia 1800, Siemens Healthcare GmbH, Henkestr, Germany). The adequacy of HD was calculated as the fractional clearance index for urea (Kt/V) and the urea reduction ratio, using the single compartment dialysis urea kinetic model. The levels of intact parathyroid hormone were measured by a commercially available enzyme-linked immunosorbent assay (Diagnostic Systems Laboratories, Webster, TX, USA).

### 5.3. Determination of Serum P-Cresyl Sulfate Levels

A Waters e2695 HPLC system that comprised a mass spectrometer (ACQUITY QDa, Waters Corporation, Milford, MA, USA) was used in this study [26]. The analytical column was a Phenomenex Luna^®^ C18 (2) (5 µ, 250 × 4.60 mm, 100 Å) with the following settings: column temperature 40 °C, flow 0.8 mL/min, and 30-µL injection. A binary gradient was applied on the mobile phase: the initial composition (95% (A) water with 0.1% formic acid/5% (B) methanol with 0.1% formic acid) was kept constant for 1 min; solvent B was then increased linearly up to 70% over 12 min and was kept constant for 2 min. For column equilibration, solvent B was reduced to 50% over 1 min and was kept constant for 2 min.

The liquid chromatography–mass spectrometry (LC–MS) gradient condition was modified as the pretreated samples were synchronously assessed in positive- or negative-ion (i.e., PCs) mode electrospray ionization. The instrument settings were as follows: desolvation temperature 600 °C, capillary voltage 0.8 kV, and sample cone 15.0 V. The mass spectrometer was operated in full scan at 50 to 450 *m*/*z* for positive-ion mode and 100 to 350 *m*/*z* for negative-ion mode. The single ion recording mode was used to monitor the individual masses of each compound (PCs: 187.0 *m*/*z*). Empower^®^ 3.0 software (Waters Corporation, Milford, MA, USA) was used for data acquisition and processing. The retention time for PCs was approximately 16.56 min. Endogenous compounds were quantified by measuring and comparing the peak areas with the calibration curve obtained from standard solutions. All the determination coefficients of linearity (r^2^) were more than 0.995. LC–MS single ion recording mode was used for single ion analysis.

### 5.4. Carotid–Femoral PWV Measurements

An applanation tonometer (SphygmoCor system, AtCor Medical, Sydney, Australia) was applied to measure the carotid–femoral pulse wave velocity (cfPWV), as previously reported [27]. After resting in a supine position for at least 10 min, patients underwent cfPWV recordings concurrent with an electrocardiogram as a timing reference for the R wave signal. The carotid-femoral distance was obtained by subtracting the carotid measurement site to sternal notch distance from the sternal notch to femoral measurement site distance. Recordings of the successive pulse waves from the carotid and femoral arteries were measured. Using integral software, which contained indices of quality to assure consistency of data on a beat-to-beat basis, the data on pulse wave and electrocardiogram were used to compute the mean interval between the pulse wave and the R wave within an average of 10 cardiac cycles. The carotid–femoral distance was obtained by subtracting the distance of the carotid measurement site to the sternal notch from the distance of the sternal notch to the femoral measurement site. Thereafter, the elapsed time and the difference in distance between the carotid and femoral arteries were used to calculate cfPWV. On the basis of the European Society of Cardiology and the European Society of Hypertension Guidelines [28], patients were sorted according to the cfPWV into the high central AS (>10 m/s) or control (≤10 m/s) group.

### 5.5. Statistical Analysis

The Kolmogorov–Smirnov test was used to examine the normality of distribution of continuous variables, which were expressed as mean ± standard deviation or as median with IQR. Comparisons between the high AS and control groups were analyzed by the independent Student’s t-test or two-tailed Mann–Whitney U test, as appropriate. Categorical data were represented as number and percentage and were analyzed using the χ^2^ test. Continuous variables that did not have a normal distribution were logarithmically transformed for use in the linear regression analysis. Multivariate logistic and linear regression analyses were used to assess the risk factors for high central AS and the relationship between all variables and cfPWV, respectively. The best cutoff PC level to predict high central AS was determined using the ROC curve to calculate the area under the curve (AUC). An analysis was regarded as significant if the *p* value was <0.05. SPSS for Windows (version 19.0; SPSS Inc., Chicago, IL, USA) was used for analyses.

## Figures and Tables

**Figure 1 toxins-12-00010-f001:**
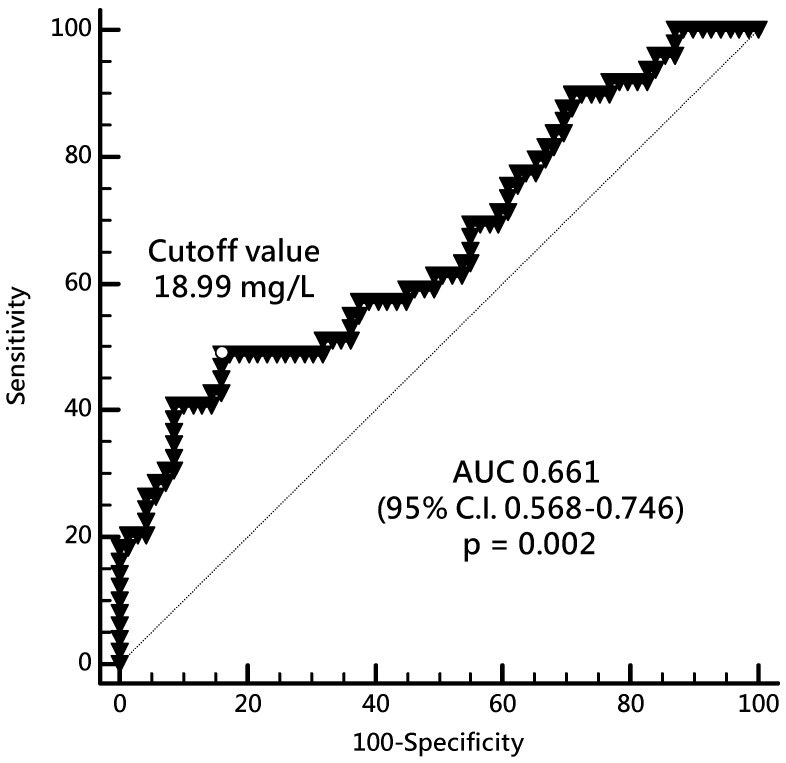
Receiver-operating characteristic curve and the *p*-cresyl sulfate cutoff level that predicts arterial stiffness in HD patients. AUC, area under the curve; CI, confidence interval; HD, hemodialysis.

**Table 1 toxins-12-00010-t001:** Clinical variables of the 118 hemodialysis patients with high and low arterial stiffness.

Characteristics	All Patients (*n* = 118)	Control Group (*n* = 69)	High Arterial Stiffness Group (*n* = 49)	*p*
Carotid–femoral PWV (m/s)	9.63 ± 2.55	7.83 ± 1.30	12.16 ± 1.48	<0.001 *
Age (years)	63.05 ± 13.28	61.32 ± 13.74	65.49 ± 12.34	0.093
Female, *n* (%)	59 (50.0)	36 (52.2)	23 (46.9)	0.575
Body mass index (kg/m^2^)	24.92 ± 5.13	24.92 ± 5.45	24.92 ± 4.69	0.994
Hemodialysis duration (months)	56.02 (24.60–111.45)	80.40 (22.38–133.80)	45.72 (26.22–74.50)	0.069
Diabetes mellitus, *n* (%)	55 (46.6)	19 (27.5)	36 (73.5)	<0.001 *
Hypertension, *n* (%)	66 (55.9)	34 (49.3)	32 (65.3)	0.084
Systolic blood pressure (mmHg)	142.13 ± 25.64	138.07 ± 26.99	147.84 ± 22.38	0.021 *
Diastolic blood pressure (mmHg)	76.02 ± 15.61	76.17 ± 16.29	75.29 ± 15.17	0.765
Heart rate (beats per minute)	75.37 ± 12.79	76 ± 13.17	74.49 ± 12.32	0.629
Blood urea nitrogen (mg/dL)	60.36 ± 14.75	59.77 ± 13.64	61.20 ± 16.31	0.604
Creatinine (mg/dL)	9.17 ± 1.99	9.47 ± 1.98	8.73 ± 1.94	0.047 *
Urea reduction rate	0.74 ± 0.04	0.74 ± 0.05	0.73 ± 0.04	0.501
Kt/V (Gotch)	1.35 ± 0.17	1.36 ± 0.19	1.33 ± 0.16	0.396
Total cholesterol (mg/dL)	143.19 ± 35.25	146.35 ± 38.11	138.73 ± 30.60	0.244
Triglyceride (mg/dL)	113.00 (86.75–178.75)	109.00 (86.50–199.00)	121.00 (85.00–174.50)	0.785
Glucose (mg/dL)	136.50 (113.75–177.00)	132.00 (110.50–162.00)	143.00 (119.50–206.00)	0.042 *
Total calcium (mg/dL)	8.96 ± 0.75	8.94 ± 0.76	8.99 ± 0.75	0.732
Phosphorus (mg/dL)	4.65 ± 1.32	4.69 ± 1.35	4.61 ± 1.31	0.746
Intact parathyroid hormone (pg/mL)	186.50 (66.60–353.35)	211.70 (101.10–413.15)	136.80 (44.40–281.75)	0.098
C-reactive protein (mg/dL)	0.34 (0.09–0.95)	0.24 (0.08–0.86)	0.57 (0.13–1.09)	0.029 *
Total *p*-cresyl sulfate (mg/L)	16.57 ± 9.07	13.95 ± 5.93	20.26 ± 11.27	<0.001 *
Angiotensin receptor blocker, *n* (%)	35 (29.7)	19 (27.)	16 (32.7)	0.549
β-blocker, *n* (%)	39 (33.1)	22 (31.9)	17 (34.7)	0.749
Calcium channel blocker, *n* (%)	46 (39.0)	29 (42.0)	17 (34.7)	0.421
Statin, *n* (%)	19 (16.1)	8 (11.6)	11(22.4)	0.114
Fibrate, *n* (%)	13 (11.0)	9 (13.0)	4 (8.2)	0.404

Values for continuous variables are shown as mean ± standard deviation or as median and interquartile range, after analysis by Student’s *t*-test or Mann–Whitney U test and according to the normality of distribution. Values that are presented as number (%) were analyzed by the chi-square test. * *p* < 0.05.

**Table 2 toxins-12-00010-t002:** Multivariate logistic regression analysis of the factors that correlated with arterial stiffness in 118 hemodialysis patients.

Variables	Odds Ratio	95% Confidence Interval	*p*
Presence of diabetes mellitus	4.095	1.429–11.739	0.009 *
Total *p*-cresyl sulfate, 1 mg/L	1.072	1.002–1.147	0.043 *
Age, 1 year	1.026	0.987–1.066	0.191
Sex (female)	0.610	0.223–1.690	0.336
C-reactive protein, 0.1 mg/dL	1.822	0.831–3.997	0.134
Glucose, 1 mg/dL	1.004	0.997–1.011	0.286
Creatinine, 1 mg/dL	0.945	0.710–1.259	0.700
Systolic blood pressure, 1 mmHg	1.002	0.983–1.022	0.838
Heart rate, 1 beat per minute	0.991	0.954–1.030	0.657
Hemodialysis duration, 1 month	0.997	0.988–1.005	0.433

Data analysis was done using the multivariate logistic regression analysis, which was adjusted for the following factors: age, sex, diabetes mellitus, systolic blood pressure, heart rate, hemodialysis duration, C-reactive protein, glucose, creatinine, and total *p*-cresyl sulfate. * *p* < 0.05.

**Table 3 toxins-12-00010-t003:** Correlation between central PWV levels and the clinical variables among 118 HD patients.

Variables	Central PWV (m/s)				
	Simple Regression		Multivariate Regression		
	r	*p*	Beta	Adjusted R^2^ Change	*p*
Age (years)	0.078	0.402	-	-	-
Female sex	−0.085	0.357	-		
Body mass index (kg/m^2^)	0.056	0.545	-	-	-
Log-HD duration (months)	−0.255	0.005 *	-	-	-
Diabetes mellitus	0.538	<0.001 *	0.446	0.283	<0.001 *
Hypertension	0.081	0.381	-	-	-
Systolic blood pressure (mmHg)	0.263	0.004 *	-	-	-
Diastolic blood pressure (mmHg)	0.055	0.551	-	-	-
Heart rate (beats per minute)	−0.112	0.228			
Blood urea nitrogen (mg/dL)	0.016	0.867	-	-	-
Creatinine (mg/dL)	−0.094	0.313	-	-	-
Urea reduction rate	−0.099	0.285	-	-	-
Kt/V (Gotch)	−0.104	0.264	-	-	-
Total cholesterol (mg/dL)	−0.068	0.462	-	-	-
Log-triglyceride (mg/dL)	−0.015	0.872	-	-	-
Log-glucose (mg/dL)	0.198	0.031 *	-	-	-
Total calcium (mg/dL)	0.028	0.763	-	-	-
Phosphorus (mg/dL)	0.065	0.487	-	-	-
Log-iPTH (pg/mL)	−0.088	0.341	-	-	-
Log-CRP (mg/dL)	0.135	0.144	-	-	-
Total *p*-cresyl sulfate (mg/L)	0.382	<0.001 *	0.174	0.028	0.018 *

Data on HD duration, triglyceride, glucose, iPTH, and CRP levels showed skewed distributions and, therefore, were log-transformed before analysis. Data analysis was done using univariate linear regression analyses or multivariate stepwise linear regression analysis adjusted for the following factors: diabetes mellitus, log-HD duration, systolic blood pressure, log-glucose, and total *p*-cresyl sulfate. * *p* < 0.05.

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
