# Peer review of "Serum P-Cresyl Sulfate Is a Predictor of Central Arterial Stiffness in Patients on Maintenance Hemodialysis"

_toxins, 2019, doi:10.3390/toxins12010010_

Round 1

Reviewer 1 Report

please explain why did you not included age and lenght of HD in the multivariate ligistic regression analysis ? please explain why did you not discuss about 25 OH D impact on AS, since there are numerous data suggestion the role of 25 OHD as an independent risk factor of endothelial dysfunction 

Author Response

Comments and Suggestions for Authors

please explain why did you not included age and lenght of HD in the multivariate ligistic regression analysis ? please explain why did you not discuss about 25 OH D impact on AS, since there are numerous data suggestion the role of 25 OHD as an independent risk factor of endothelial dysfunction 

Ans: Thanks for your comments

Evidence had shown that increasing age and length of HD of patients were risk factors for development of arterial stiffness1,2. Because there were no significant difference of age and duration of HD between the high arterial stiffness and control group shown in Table 1, we did not use variables without significant difference between these two groups for further multivariate logistic regression analysis for arterial stiffness. However, if we used age and HD duration in the multivariate logistic regression analysis, it again showed that age (OR 1.026, 95% C.I. 0.987-1.066, p = 0.191) and HD duration (OR 0.997, 95% 0.988-1.005, p = 0.433) were not significant risk factors for development of arterial stiffness.

       In HD patients, Meijers et al. showed that p-cresyl sufate, serum phosphorus and calcium were significantly associated with endothelial dysfunction, whereas treatment with active vitamin D was inversely associated with, which highlighted the role of p-resyl sulfate and CKD-MBD2. In this study we did not examine the serum level of 25 Vit D but did find that there was no significant role of calcium, phosphorus and iPTH related to arterial stiffness or carotid-femoral pulse wave velocity. Nevertheless, we find that p-cresyl sulfate was a significant risk factor the development of arterial stiffness. 

Meijers, B.K.; Van Kerckhoven, S.; Verbeke, K.; Dehaen, W.; Vanrenterghem, Y.; Hoylaerts, M.F.; Evenepoel, P. The uremic retention solute p-cresyl sulfate and markers of endothelial damage. American journal of kidney diseases : the official journal of the National Kidney Foundation 2009, 54, 891-901.

Cecelja, M.; Chowienczyk, P. Dissociation of aortic pulse wave velocity with risk factors for cardiovascular disease other than hypertension: A systematic review. Hypertension 2009, 54, 1328-1336

Reviewer 2 Report

How was the carotid-femoral distance measured in the PWV assessment?

Do the authors have elements concerning the calcification processes in these dialysis patients? It would be interesting to assess whether the association between p-cresyl sulfate levels and PWV is mediated by phenomena of arterial calcification.

Heart rate must be added to the analysis (including tables). The results concerning the relationship between PWV and p-cresyl sulfate should be re-analyzed and PWV values should be adjusted for age, sex, presence of diabetes, heart rate and systolic blood pressure.

The English language could be greatly improved.

Author Response

Comments and Suggestions for Authors

How was the carotid-femoral distance measured in the PWV assessment?

Ans: Thanks for your comments. We would add the description of carotid-femoral distance as “The carotid-femoral distance was obtained by subtracting the carotid measurement site to sternal notch distance from the sternal notch to femoral measurement site distance”.  

Do the authors have elements concerning the calcification processes in these dialysis patients? It would be interesting to assess whether the association between p-cresyl sulfate levels and PWV is mediated by phenomena of arterial calcification.

Ans: Thanks for your comments. In this study, we analyzed the correlation between laboratory, clinical, demographic variables and cfPWV and found that only p-cresyl sulfate and DM but not serum calcium, phosphorus or iPTH correlated significantly with cfPWV (Table 3). We further analyzed the correlation between p-cresyl sulfate and calcium, phosphorus and iPTH according to your comments and revealed no significant correlation (r = 0.07, 0.00, -0.096, p = 0.453, 0.999, 0.299, respectively; data not shown).

Heart rate must be added to the analysis (including tables). The results concerning the relationship between PWV and p-cresyl sulfate should be re-analyzed and PWV values should be adjusted for age, sex, presence of diabetes, heart rate and systolic blood pressure.

Ans: Thanks for your comments. We re-analyzed by including the variables according to your suggestions and showed the data in Table 2. P-cresyl sulfate and DM remained as the significant risk factors for the developments of central arterial stiffness in HD patients.

The English language could be greatly improved.

Ans: Thanks for your comments. We will have this manuscript edited by Enago (INQ-8154930519) (www.enago.tw) before this submission.

Reviewer 3 Report

Please refer to the uploaded files

Author Response

General suggestions:

Numerals should be written in Arabic or words; format should be uniform. Like line 9 “Forty-nine (41.5) patients…” and Line 47-48 “Fifty-five (46.6%) and 66 (55.9%) patients…”, etc.

Ans: Thanks for your comments. We will revise the whole manuscript according to your suggestions.

The “Table 1” should be aligned with the following tables, it’s wider than other tables. (problems with typesetting?)

Ans: Thanks for your comments. We re-aligned the Tables according to your suggestions.

Table legends should be corresponded to the asterisk in the table. It’s hard to read without any directivity.

Ans: Thanks for your comments. We revised the description of table legend according to your suggestions.

Line 82-86, these two paragraphs are table legend or text? Please confirm.

Ans: Thanks for your comments. These two paragraphs are table legend.

Line 10, “had higher incidence” should be “had a higher incidence”;

Line 13, “increased risk at developing” should be “increased risk in developing”;
Line 27, “had showed that” should be “had shown that”;
Line 38 and Line 126, “in vitro” should be italic;
Line 43, “carotid-femoral cfPWV (cfPWV)” should be “carotid-femoral PWV (cfPWV)”;
Line 47, “with median” should be “with a median”;
Line 74, “Value of cfPWV” should be “The value of cfPWV”;
Line 106, “having DM lead to” should be “having DM leads to”;
Line 127, “through mechanism” should be “through the mechanism”;
Line 128, “of increasing” should be “of increased”;
Line 135, “signalling” should be “signaling”;
Line 148, “mechanism to be further elucidated” should be “mechanism needs to be further elucidated”.

Ans: Thanks for your comments. We revised these sentences according to your suggestions. In addition, we will have this manuscript edited by Enago (INQ-8154930519) (www.enago.tw)before this submission.

Round 2

Reviewer 2 Report

The authors have correctly modified the text according to my suggestions.

Reviewer 3 Report

Line 81-85, the main text format (align) not consistent with other paragraphs.

Line 88-91, the table legend format (align) is not consistent with other legends.